# [Re:] Training Binary Neural Networks using the Bayesian Learning Rule

1 ## Reproducibility Summary

2 *(1) gives a mathematically principled approach to solve the discrete optimization problem that occurs in the case of*
3 *Binary Neural Networks and claims to give a similar performance on various classification benchmarks such as MNIST,*
4 *CIFAR-10, and CIFAR-100 as compared to their full-precision counterparts, as well as other recent algorithms to*
5 *train BNNs like PMF and Bop. The paper also claims that the BayesBiNN method has an application in the continual*
6 *learning domain as it helps in overcoming catastrophic forgetting of the past by using the posterior approximation of*
7 *the previous task as a prior for the upcoming task. We try to reproduce all the results presented in the original paper by*
8 *making a separate and independent codebase.*

9 **Scope of Reproducibility**

10 We try to verify the performance of our re-implementation of the BayesBiNN optimizer on various classification and
11 regression benchmarks. We also implemented the STE optimizer which was the central baseline model used in the
12 paper. Finally, we tried to evaluate the results of BayesBiNN on the continual learning benchmark to get a better insight.

13 **Methodology**

14 We developed our separate code-base, consisting of an end-to-end trainer with a Keras-like interface, for the reproduction
15 which includes the implementation of the BayesBiNN and STE optimizer. We did refer to the author's code open-sourced
16 on GitHub to get some insights about the hyperparameters and other doubts that emerged during code development.

17 **Results**

18 We reproduced the accuracy of the BayesBiNN optimizer within less than 0.5% of the originally reported value, which
19 upholds the conclusion that it performs nearly as well as its full-precision counterpart in classification tasks. When we
20 tried this in a semantic segmentation context, we found that the results were very underwhelming and in contrast with
21 the seemingly good results by the STE optimizer even with much hyperparameter tuning. We can conclude that, like
22 other Bayesian methods, it is difficult to train BayesBiNN on more complex tasks.

23 **What was easy**

24 After we worked out the mathematics behind the BayesBiNN approach, we developed a pseudo-code for the optimization
25 process which along with references from the author's code, helped us a lot in our reproduction study.

26 **What was difficult**

27 Some of the hyperparameters were not mentioned by the authors in their paper so it was difficult to approximate the
28 values of those parameters. The lack of resources was the next big difficulty that we faced.

29 **Communication with original authors**

30 We had a very fruitful conversation with the authors, which helped us in better understanding the BayesBiNN approach
31 and its extension to the segmentation domain. The detailed pointers are given at the end of this report.

## 1 Introduction

Deep Learning is moving towards larger and larger parameters day-by-day, which often makes it difficult to run on resource-constraint devices like mobile phones. Binary Neural Networks (BNNs) could act as a savior in such situations, helping in largely saving storage and computational costs. The problem of optimizing this binary set of weights is clearly a discrete optimization problem. Previous approaches like Straight-Through Estimator (STE) and Binary Optimizer (Bop) tend to ignore this and use gradient-based methods, which still worked in practice. The paper presents a mathematically principled approach for training BNNs which also justifies the current approaches.

## 2 Scope of reproducibility

The paper mentions a bayesian approach to solve the discrete optimization problem in the case of Binary Neural Networks (BNNs). The outcome of this approach was a BayesBiNN optimizer which could be used to train BNNs and achieve similar accuracy as compared to their full-precision counterparts. To verify the claims given in the paper, we target to achieve the following objectives:

- Work out and present the mathematics behind BayesBiNN in a simpler way and present the pseudo-code to the optimizer.
- Implement the BayesBiNN optimizer and STE optimizer to verify the accuracy on tasks of varying complexities, as reported in the original paper.
- Reproduce the results for other baselines present in the paper such as proximal mean-field (PMF) according to the hyper-parameters given in the paper.
- Evaluating the performance of BayesBiNN optimizer in more complex domains like semantic segmentation.

## 3 Methodology

We have re-implemented the algorithm proposed in the paper from scratch using PyTorch and created an end-to-end model trainer with a Keras-like interface. We referred to the code given by the authors for the baseline model hyperparameters and the source of synthetic datasets. Following is the algorithmic form of what the authors have presented in the paper.

---

**Algorithm 1:** Bayesian Learning rule for BayesBiNN

---

**Input:** Initialize $\lambda$
**for** *number of training epochs* **do**
  **for** *i = 1,...,number of mini-batch examples* **do**
    Sample $\epsilon \sim \mathcal{U}(0,1)$ and set $\delta = \frac{1}{2}\log\frac{\epsilon}{1-\epsilon}$
    Initialize $w_b = \tanh((\lambda + \delta)/\tau)$
    Compute following using *gumbel-softmax* trick

$$g_i := \frac{1}{M}\nabla_{w_b}l(y_i, f_{w_r}(x_i))$$

$$s_i := \frac{N(1 - w_b^2)}{\tau(1 - \tanh(\lambda)^2)}$$

  **end**
  Update $\mu$ and $\lambda$ using following equation
$$\mu \leftarrow \tanh(\lambda)$$
$$\lambda \leftarrow (1-\alpha)\lambda - \alpha[\sum_{i=1}^{M}(s_i \odot g_i) - \lambda_0]$$

**end**

---

This would make the paper more interpretive in terms of implementation.

Some of the mathematical expressions mentioned in the original paper were presented from various sources and missed out several intermediate steps which we found to be very important while reproducing the paper from scratch. Here we present a step-wise derivation of some important expressions written in the original paper:

Bayesian formulation of the discrete optimization problem, in which loss has to be minimized w.r.t posterior $q(w)$, given prior $p(w)$ can be written as:

$$\mathbb{E}_{q(w)}[\sum_{i=1}^{N} l(y_i, f_w(x_i))] + \mathcal{D}_{KL}[q(w)||p(w)]$$

To solve the above optimization problem, Bayesian learning rule given in (6) is applied, assuming solution to be a part of minimal exponential family of distribution, given by:

$$q(w) = h(w)exp[\lambda^T \phi(w) - A(\lambda)]$$

where base measure $h(w)$ is assumed to be 1. Following is the update rule used to learn $\lambda$:

$$\lambda \leftarrow (1 - \rho)\lambda - \rho[\nabla_\mu \mathbb{E}_{q(w)}[l(y_i f_w(x_i))] - \lambda_0]$$

where $\rho$ is the learning rate, $\mu = \mathbb{E}_{q(w)}[\phi(w)]$. Bernoulli distribution being a special case of minimal exponential family distribution, we assume prior $p(w) \sim \text{Bern}(p)$ with $p = 0.5$, and posterior $q(w)$ to be mean-field bernoulli distribution:

$$q(w) = \prod_{j=1}^{W} p_j^{\frac{1+w_j}{2}} (1 - p_j)^{\frac{1-w_j}{2}}$$

For weight $j$,

$$q(w_j) = \exp(\frac{1}{2}(1 + w_j)\log p_j + \frac{1}{2}(1 - w_j)\log(1 - p_j))$$

$$= \exp(\underbrace{w_j}_{\phi(w)} \underbrace{\frac{1}{2}\log \frac{p}{1-p}}_{\lambda}) + \frac{1}{2}\log(p(1 - p))$$

Comparing above expression with minimal exponential family distribution, we can say:

$$\lambda = \frac{1}{2}\log \frac{p}{1-p} \text{ and } \phi(w) = w.$$

We defined $\mu = \mathbb{E}_{q(w)}[\phi(w)]$,

$$\mu = \int wq(w)dw = \mathbb{E}[q(w)] = \sum_{w^i \in \{-1,1\}} w^i q(w^i)$$

$$= \sum_{w^i \in \{-1,1\}} w^i p^{\frac{1+w^i}{2}}(1 - p)^{\frac{1-w^i}{2}} = -(1 - p) + p$$

$$= 2p - 1$$

From above derivations we can say that, $p = 1/(1 + \exp(-2\lambda)) = \text{Sigmoid}(2\lambda)$ and $q(w) \sim \text{Bern}(p)$.

To implement the update rule, we need to compute the gradient with respect to $\mu$. Original paper uses a reparamaterization trick called Gumbel-softmax trick (7), which is used to relax the discrete random variables of a concrete distribution (for eg, bernoulli distribution). Binary concrete relaxation (7) of Binary concrete random variable $X \in (0, 1)$ with distribution $X \sim \text{BinConcrete}(\alpha, \lambda)$ with temperature $\lambda$ and location $\alpha$,

$$X = \frac{1}{1 + \exp(-(\log \alpha + L)/\lambda)}$$

where $L \sim \text{Logistic}$. And its density is given by

$$p_{\alpha,\lambda}(x) = \frac{\lambda \alpha x^{-\lambda-1}(1 - x)^{-\lambda-1}}{(\alpha x^{-\lambda} + (1 - x)^{-\lambda})^2}$$

Using above expressions, for binary weights $w_j \in \{0, 1\}$, relaxed variable $w_r^{\epsilon_j, \tau}(p_j) \in (0, 1)$ can be used with temperature $\tau$ and $\alpha = e^{2\lambda}$ given by

$$w_r^{\epsilon_j, \tau}(p_j) = \frac{1}{1 + \exp(-\frac{2\lambda_j + 2\delta_j}{\tau})},$$

where $\delta_j \sim \text{Logistic}$ and its density is given by

$$p(w_r^{\epsilon_j, \tau}(p_j)) = \frac{\tau e^{2\lambda} w_r^{\epsilon_j, \tau}(p_j)^{-\tau-1}(1 - w_r^{\epsilon_j, \tau}(p_j))^{-\tau-1}}{(e^{2\lambda} w_r^{\epsilon_j, \tau}(p_j)^{-\tau} + (1 - w_r^{\epsilon_j, \tau}(p_j))^{-\tau})^2}$$

## 4 Experimental setup

### 4.1 Model descriptions

We kept the model architectures the same as mentioned in the original paper to maintain uniformity and implemented them ourselves. For the MNIST classification task, we used the BinaryConnect architecture and for the CIFAR classification task, we use the VGGBinaryConnect architecture. The authors also compared their BayesBiNN method with the LR-Net method in (8). We implemented the same model architecture as in the LR-Net paper. The detailed architectures are mentioned in the supplementary material provided with this report. For the segmentation task, we used the original U-Net architecture, mentioned in (11) with a minor difference that we introduced a BatchNorm layer after every convolution layer.

### 4.2 Datasets

The datasets used for image classification tasks are MNIST, CIFAR-10, and CIFAR-100. For generating visualizations for the BayesBiNN and STE methods, we used small toy datasets, the Snelson dataset (10) for regression problems, and Two Moon's dataset (9) for classification problems. For the segmentation part, we used the Brain Tissue segmentation dataset taken from (11), and for the continual learning visualizations, we used the permuted MNIST dataset (12). The pre-processing of inputs has been kept the same as mentioned in the original paper and has been detailed below.

**Pre-processing**: For the MNIST dataset we simply normalize the images and do not perform data augmentation. We keep our validation split as 0.1 uniformly across all sets of experiments except the comparison with the LR-Net method (8). For the CIFAR datasets also, we perform the normalization of images along with data-augmentation where we generate images by randomly cropping a 32x32 image from a 40x40 padded image. Finally, for our semantic segmentation task, we had a very small dataset of 30 images, out of which 24 were chosen for training and 6 for obtaining the validation score. No other pre-processing has been done in this case.

### 4.3 Hyperparameters

We have used the hyper-parameters given in the original paper. Table 1 contains the list of all the parameters we used for our experiments:

| Optimizer | Parameter | MNIST | CIFAR10 | CIFAR100 | Snelson Dataset | 2 Moons Dataset |
|---|---|---|---|---|---|---|
| BayesBiNN | MC steps | 1 | 1 | 1 | 1 | 5 |
| | Initial LR | $10^{-4}$ | $3.10^{-4}$ | $3.10^{-4}$ | $10^{-4}$ | $10^{-3}$ |
| | Final LR | $10^{-16}$ | $10^{-16}$ | $10^{-16}$ | $10^{-5}$ | $10^{-5}$ |
| | LR Scheduler | Cosine | Cosine | Cosine | MultiStepLR | MultiStepLR |
| | Temperature $\tau$ | $10^{-10}$ | $10^{-10}$ | $10^{-8}$ | 1 | 1 |
| | Initialization $\lambda$ | $\pm 10$ | $\pm 10$ | $\pm 10$ | $\pm 10$ | $\pm 15$ |
| STE | Initial LR | $10^{-2}$ | $10^{-2}$ | $10^{-2}$ | $10^{-1}$ | $10^{-1}$ |
| | Final LR | $10^{-16}$ | $10^{-16}$ | $10^{-16}$ | $10^{-1}$ | $10^{-3}$ |
| | LR Scheduler | Cosine | Cosine | Cosine | MultiStepLR | MultiStepLR |
| Adam (Full Precision) | Initial LR | $10^{-5}$ | $10^{-4}$ | $10^{-4}$ | - | - |
| | Final LR | Step | Step | Step | - | - |
| | LR Scheduler | 1 | 100 | 100 | - | - |

Table 1: Training setting for different optimizers on MNIST, CIFAR10, and CIFAR100 datasets.

### 4.4 Computational requirements

All our final experimental results were performed on a machine having 1 NVIDIA Tesla V100 GPU and 1 single-core system with 16 GB memory. Training the Binary Network with BayesBiNN optimizer for a single run, takes around 2.5 GPU hours for MNIST, 5.5 GPU hours for CIFAR-10, and around 8.5 GPU hours for the CIFAR-100 dataset, in the current experimental setup.

## 5 Results

In Table 1 we report our results for various classification benchmarks using our implemented BayesBiNN and STE optimizer. We notice that we get a difference of less than 0.1% as compared to that in the original paper. We generated

the results for baseline STE optimizer and full-precision networks by evaluating our implementation of these methods. We also generated the results of PMF, by modifying its original open-sourced code and using the hyperparameters mentioned in the original paper.

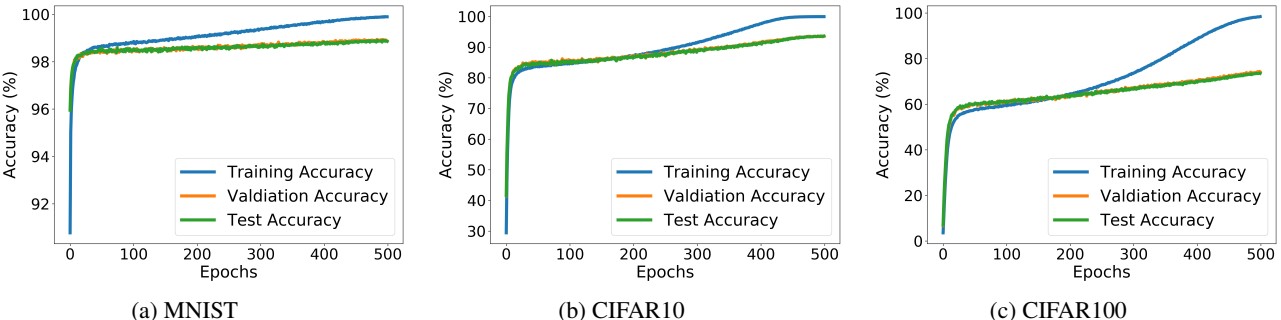

Figure 1: Training/Validation/Test accuracy using BayesBiNN optimizer

| Datasets | Optimizer | Training Accuracy | Validation Accuracy | Test Accuracy |
|---|---|---|---|---|
| MNIST | BayesBiNN(ours) | 99.90 ±0.01% | 99.89 ±0.07% | 98.87 ±0.06% |
| | BayesBiNN(orig.) | 99.85 ±0.05% | 99.02 ±0.13% | 98.86 ±0.05% |
| | STE | 99.90 ±0.01% | 98.86 ±0.09% | 98.89 ±0.05% |
| | PMF | - | 98.73% | - |
| | Adam (Full Precision) | 99.98 ±0.01% | 99.02 ±0.04% | 99.02 ±0.01% |
| CIFAR10 | BayesBiNN(ours) | 99.96 ±0.01% | 93.59 ±0.45% | 93.54 ±0.26% |
| | BayesBiNN(orig.) | 99.96 ±0.01% | 94.23 ±0.41% | 93.72 ±0.16% |
| | STE | 99.99 ±0.01% | 93.77 ±0.06% | 93.54 ±0.08% |
| | PMF | - | 91.98% | - |
| | Adam (Full Precision) | 99.99 ±0.01% | 94.27 ±0.15% | 94.38 ±0.16% |
| CIFAR100 | BayesBiNN(ours) | 98.35 ±0.1% | 74.13 ±0.78% | 73.56 ±0.06% |
| | BayesBiNN(orig.) | 98.02 ±0.18% | 74.76 ±0.41% | 73.68 ±0.31% |
| | STE | 99.22 ±0.03% | 72.74 ±0.06% | 73.25 ±0.26% |
| | PMF | - | 70.82% | - |
| | Adam (Full Precision) | 99.89 ±0.02% | 75.04 ±0.71% | 74.80 ±0.39% |

Table 2: Results of different optimizers trained on MNIST, CIFAR10, and CIFAR100.

## 5.1 Comparison with LR-Net

Authors compared their BayesBiNN approach to the LR-Net method presented in (8). We tried to reproduce the result for the same setting. In this comparison, the data pre-processing and augmentation methods remain the same as mentioned in section 4.2, but we do not split the data in training and validation sets in this case. We denote the test accuracies after 190 epochs in the case of MNIST and 290 epochs in the case of CIFAR-10, as done in the original paper to maintain uniformity. Note that, our accuracy is matching with that of the original authors in the case of MNIST but not in the case of CIFAR-10. We suspect that this is due to some difference in Batch-Norm layers used.

| Optimizer | MNIST | CIFAR10 |
|---|---|---|
| BayesBiNN (ours) | **99.52%** | 84.49% |
| BayesBiNN (orig.) | 99.50% | **93.97%** |
| LR-net (8) | 99.47% | 93.18% |

Table 3: Test accuracy of BayesBiNN and LRNet.

## 5.2 Continual Learning

As mentioned in the original paper, we try to reproduce the author's claims about weight distribution across tasks in a simple continual learning domain tested on Permuted MNIST. As we can see, as we learn across the tasks, the curve

becomes flat from the middle conveying that the weights become more deterministic. Our result matches with the claims in the original paper.

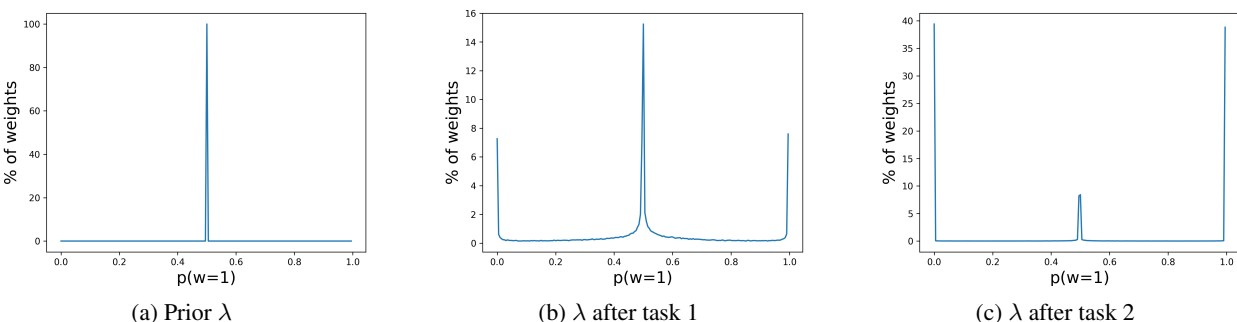

(a) Prior $\lambda$  |  (b) $\lambda$ after task 1  |  (c) $\lambda$ after task 2

Figure 2: Distribution of $p(w = 1)$ across consecutive learning tasks

## 5.3 Visualization using Synthetic Dataset

In the original paper, the authors present visualizations on binary classification (Two moons dataset(9)) and toy regression (Snelson dataset(10)) using STE and BayesBiNN optimizer. For the classification task, the authors claimed that STE is a more deterministic classifier compared to BayesBiNN. We reproduced this experiment and the results depicted in Figure 3 seem to be consistent with the author's claim. For the regression task, we conclude that the author's claim about BayesBiNN (mean) giving a smoother curve compared to STE is true, which can also be seen in Figure 4.

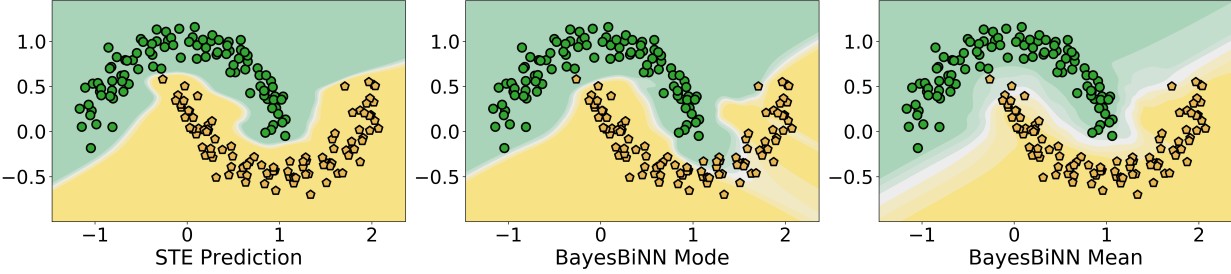

Figure 3: Classification on Two Moons dataset using STE and BayesBiNN optimizer.

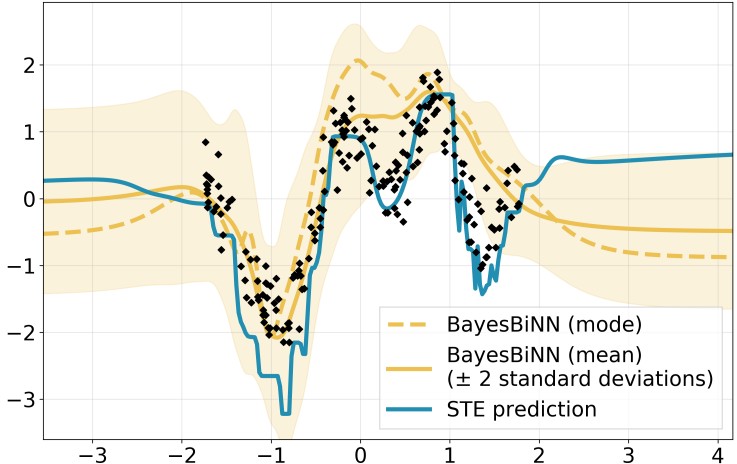

Figure 4: Regression on Snelson dataset using STE and BayesBiNN optimizer.

| Temperature | 10 | 1 | 0.1 | $10^{-2}$ | $10^{-3}$ | $10^{-4}$ |
|---|---|---|---|---|---|---|
| MSE Loss | 1.313 | 0.208 | 2.151 | 0.443 | 0.231 | 0.199 |
| Temperature | $10^{-5}$ | $10^{-6}$ | $10^{-7}$ | $10^{-8}$ | $10^{-9}$ | $10^{-10}$ |
| MSE Loss | 0.156 | 0.127 | 0.173 | 0.122 | 0.195 | 0.173 |

Table 4: Mean square error loss of Snelson dataset for different temperatures.

## 5.4 Extended Results (Semantic Segmentation)

We tried to validate the performance of the BayesBiNN optimizer on more complex tasks like Semantic Segmentation. Unfortunately, the results with BayesBiNN were quite underwhelming as compared to STE and its full-precision counterpart. We tried various parameters to improve its performance but none seemed to work. We had a brief discussion with the authors regarding this issue and the authors suggested that Bayesian models are intrinsically very difficult to train. For the results shown in Table 5 and Figure 5, we have used the hyperparameters denoted in Table 1.

| | BayesBiNN | STE | Adam (Full Precision) |
|---|---|---|---|
| Validation Score | 0.4102 | 0.3108 | 0.2943 |

Table 5: (1 - IoU) score for validation set

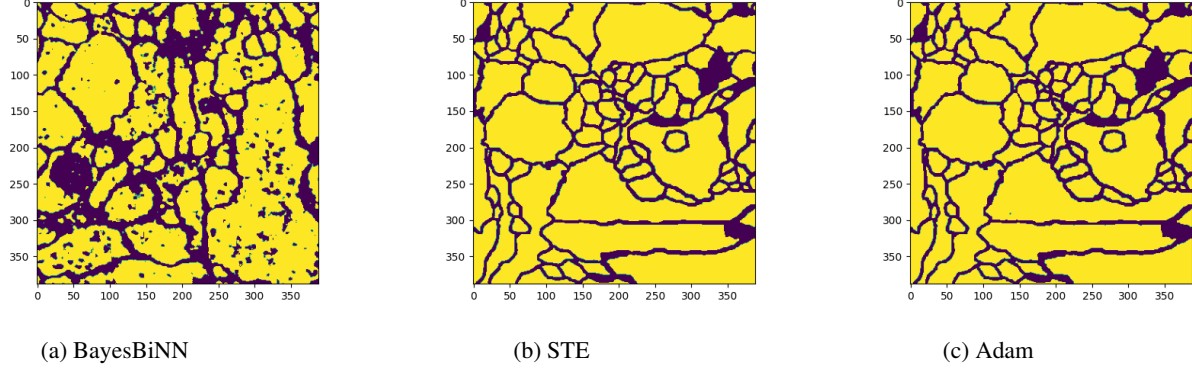

(a) BayesBiNN        (b) STE        (c) Adam

Figure 5: Some samples of segmented image outputs

## 6 Discussion

We reproduced almost all the experiments given in the original paper and most of our results match with the original claims. While this BayesBiNN approach is mathematically principled, we tried to take a step forward, by using that optimizer on a single segmentation task. However, the results were against our expectation and the result of segmentation was a zoomed segmented image of the input with lots of noise. Apart from this, even in the case of comparison with the LR-Net method, our accuracy differs from that of the original authors, which we feel might be due to some difference in architecture chosen. The major contribution of our work is developing a code base library based on PyTorch with a Keras type interface for training BNNs with several different methods in its arsenal. This would reduce the coding efforts while training a BNNs and could help in future research as benchmarking platform.

### 6.1 What was easy

The original paper contained a very good explanation of the mathematics behind the BayesBiNN approach. After we worked that out the pseudo-code as pointed out in Algorithm 1, the basic implementation of the optimizer became easy and easily verifiable by the author's original code. The appendix in the original paper contained a list of various hyper-parameters used for experiments. This helped us a lot while running the experiments and deciding the range of hyper-parameters while doing ablation studies.

## 6.2 What was difficult

The most difficult part here was running a large number of experiments in lack of many computational resources. This difficulty was increased since we are taking an average of 5 runs while reporting all our results. Apart from this, we also faced some difficulty in taking care of the hyper-parameters, which were not mentioned in the original paper (like momentum coefficient). To cater to that, we had to guess some possible values of the hyper-parameters and run small random searches to find a good candidate. Finally, we also faced difficulty while reproducing the results for the baselines PMF and Bop and adapting their experimental settings to match with those used in the original BayesBiNN paper. Since their code was written a long time ago and used older technologies, this task took us a lot of time.

## 6.3 Communication with original authors

We did not understand the intent of the authors for choosing temperature as 1 in the case of experiments on synthetic datasets. We were also curious about the author's view on segmentation tasks using BayesBiNN. We mailed this, along with the review of their paper, to the authors to ask for some pointers. They gave the following major pointers:

- It is reasonable that at high temperatures the learned distribution will have high variance. The mode mentioned in the paper refers to the $\text{sign}(\hat{(w)})$, where $\hat{(w)}$ denotes the expectation of the learned posterior Bernoulli distribution. It is not appropriate to directly use the continuous $\hat{(w)}$ as the mode. Another way is to use mean, which samples from the learned posterior Bernoulli distribution, and then make predictions using ensemble learning.

- STE is more stable and suggested by the authors to act as a baseline, in particular, Adam STE first, to make sure binary networks work. As shown in the paper, there is literally very little difference between STE and BayesBiNN but indeed the latter is difficult to train, as most Bayesian optimizers.

## Broader Impact

Recent researches (3) mention that training a single big transformer model could emit around 626,155 lbs $CO_2$ which is around 5 times of average carbon emission by a car in its total lifetime. Clearly, Deep Learning takes a huge toll on the environment which is why there has been an increased focus on much more energy-efficient "Green AI". BNNs intrinsically have far less computational and space complexity as compared to their full-precision counterparts and as we can see above they can also achieve accuracy close to the full-precision networks, at least in the classification tasks, and also show the potential of expanding well to more complex segmentation tasks. This can help us a lot in moving towards cleaner Deep Learning. This class of technology also provides a huge set of opportunities in extending AI to edge devices with much smaller and low-energy systems. We feel that its potential impact on the environment and sustainability is at par with its academic importance, that is why we see it as a much larger thing than just a set of publications.

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
