# OpenReview forum: "[Re:] Training Binary Neural Networks using the Bayesian Learning Rule"
_ML_Reproducibility_Challenge/2020 — RC2020_

### Official Review · AnonReviewer1 · 2021-03-01
**Successful reproduction, but report quality not good enough to be part of rescience journal**

**Rating:** 6
**Confidence:** 4

**Review:**

# Reproducibility Summary :
Complete

# Scope of reproducibility:
Clearly defined

# Code:
Re-implemented.

# Communication with original authors
Reports one communication exchange with original authors.

# Hyperparameter Search
The report mentions the need to do some random search on unmentioned hyperparameters (such as momentum coefficient). There is no clear mention of hyperparameter search, what was the search space, budget and algorithm used for the main hyperparameters however. What I understand is that they re-used the original ones.

# Ablation Study:
There is no ablation study

# Discussion on results
The reproduction of the results is well detailed and the table help understand what was reproduced and what was not.

# Recommendations for reproducibility
There is no clear recommendations.

# Results beyond the paper
They tried applying the model on a task that was not in the original paper, the semantic segmentation. The dataset was however extremely small which makes me uncertain about how insightful the results are.

# Overall organization and clarity
The overall organisation of the paper and the presentation of the results is clear. There is many mistakes (see minor comments below) that should be corrected. Section ‘Methodology’ is hard to understand and brings no value with respect to original paper. Pointing the reader to original paper to better understand the maths beyond BayesBiNN would make a better job in my opinion.

# Comments

Algorithm 1 is difficult to understand without more explanations.

The ‘Methodology’ section was difficult to follow. The explanations of the equations are too succinct, lack motivations and explanations. Looking at the original paper, I find it easier to understand the equations based on their explanations, so I am wondering what is the value of the presentation of the equations in this report.

Line 145:  However, the results were against our intuition and the result of segmentation were a zoomed segmented image of the input with lots of noise.

It’s not clear what is meant here. What was the intuition? What is the result and what should it have been? Should it not be zoomed?

# Minor comments
Line 36. STE and BOP acronyms should be introduced.

Line 100. by a randomly -> by randomly

Line 118. reporduce -> reproduce

Line 120: but the validation split is made 0. I don’t understand what it means to make a split 0.

Line 136 Semantic Semantic -> Semantic Segmentation

Line 137 it’s full-precision -> its full precision

Line 138 it’s performance -> its performance

Line 140 we have use -> we have used

Line 167 The gave -> They gave

Line 178 in it’s total -> in its total

Line 184 that it’s -> that its

**Familiar With The Original Paper:**

I have not read the original paper

**Reproducibility Summary:**

Report has summary

---

### Official Review · AnonReviewer3 · 2021-03-02
**This paper provides a detailed description and well discussion about reimplementation of BayesBNN. It also provides the results of extended task (semantic segmentation), which may help understanding the limitation of BayesBNN.**

**Rating:** 7
**Confidence:** 3

**Review:**

This paper summarizes the reproducibility of BayesBNN and gives a clear scope of reproducibility. It provides a re-implemented codebase to reproduce the results of the original paper. In addition, this paper also discusses the extended results for semantic segmentation. The paper is well-written and easy to understand. Couples questions:

1. For the case of unmatched CIFAR-10 results, I am wondering why the Batch-Norm layers may cause a large-cap of test accuracy.
2. For the semantic segmentation task, is it possible the limited size of training data may affect the model performance?
3. Line 183, what do you mean by “cleaner deep learning”?


**Familiar With The Original Paper:**

I have not read the original paper

**Reproducibility Summary:**

Report has summary

---

### Decision · Program_Chairs · 2021-03-31

**Decision:**

Accept

**Comment:**

Selected for ReScience-C Journal Publication.

This paper presents an extensive recreation of many results from the original work. In addition to being clear and well-presented, this work extends the original by evaluating on a new task: semantic segmentation. They find a negative result here, which is likely interesting to those looking to build up on the original work.